# Short-Term Intensity Prediction of Tropical Cyclones Based on Multi-Source Data Fusion with Adaptive Weight Learning

Wei Tian [1,*] , Ping Song [2], Yuanyuan Chen [1], Haifeng Xu [1], Cheng Jin [3] and Kenny Thiam Choy Lim Kam Sian [4]

1 School of Software, Nanjing University of Information Science and Technology, No. 219, Ningliu Road, Nanjing 210044, China; cyy@nuist.edu.cn (Y.C.); xuhaif@nuist.edu.cn (H.X.)
2 School of Computer, Nanjing University of Information Science and Technology, No. 219, Ningliu Road, Nanjing 210044, China; sping@nuist.edu.cn
3 Key Laboratory of Smart Earth, Beijing 100000, China; jinchengno1@163.com
4 School of Atmospheric Science and Remote Sensing, Wuxi University, 333 Xishan Avenue, Wuxi 214105, China; kennylimks@cwxu.edu.cn
* Correspondence: tw@nuist.edu.cn

**Abstract:** Tropical cyclones (TCs) can cause significant economic damage and loss of life in coastal areas. Therefore, TC prediction has become a crucial topic in current research. In recent years, TC track prediction has progressed considerably, and intensity prediction remains a challenge due to the complex mechanism of TC structure. In this study, we propose a model for short-term intensity prediction based on adaptive weight learning (AWL-Net) for the evolution of the TC's structure as well as intensity changes, exploring the multidimensional fusion of features including TC morphology, structure, and scale. Furthermore, in addition to using satellite imageries, we construct a dataset that can more comprehensively explore the degree of TC cloud organization and structure evolution. Considering the information difference between multi-source data, a multi-branch structure is constructed and adaptive weight learning (AWL) is designed. In addition, according to the three-dimensional dynamic features of TC, 3D Convolutional Gated Recurrent (3D ConvGRU) is used to achieve feature enhancement, and then 3D Convolutional Neural Network (CNN) is used to capture and learn TC temporal and spatial features. Experiments on a sample of northwest Pacific TCs and official agency TC intensity prediction records are used to validate the effectiveness of our proposed model, and the results show that our model is able to focus well on the spatial and temporal features associated with TC intensity changes, with a root mean square error (RMSE) of 10.62 kt for the TC 24 h intensity forecast.

**Keywords:** tropical cyclone; intensity prediction; remote sensing data; adaptive weight learning

## 1. Introduction

The tropical cyclone (TC) occurs oi the tropical or subtropical ocean, with sufficient ocean temperature and water vapor as the support of its development, and is a weather system with organized convection. Since the landfall of a TC is always accompanied by storms and floods, tsunamis, mudslides, etc., it destroys buildings and facilities, leading to large-scale damage, causing huge social and economic losses to coastal areas and seriously damaging human life and property safety [1–3]. For example, Hurricane Katrina struck New Orleans and its surrounding areas in the United States in August 2005. It had wind speeds reaching up to 175 miles per hour and a storm surge of 20 feet, resulting in the death of 1800 people and causing economic losses exceeding USD 125 billion. Typhoon Haiyan made landfall in the central Philippines in November 2013 with wind speeds of up to 195 miles per hour, making it one of the strongest recorded typhoons. The storm surge of more than 13 feet and brought heavy precipitation, triggering severe flash floods and mudslides. The typhoon caused over 6000 deaths and resulted in significant damage to infrastructure, farmlands, and villages. Therefore, predicting TC activity is the key to

disaster prevention and mitigation, and TC intensity and track prediction are the most important [4]. With the development of observation technology and science, TC track prediction has made much progress [5,6]. However, TC intensity prediction studies are experiencing a difficult time due to their dependence on factors such as different scales, temperatures and time, as well as the lack of knowledge about the structural variations of TCs during their development and the influence of the physical factors to which they are subjected [7,8].

To further monitor and track TCs, meteorologists use tools such as satellite observations, aircraft observations, and weather radar for analysis, with satellite observations being one of the most commonly used tools. Remote sensing satellites can provide high-resolution cloud maps, infrared images, water vapor images, etc., which facilitates researchers to better understand and learn about the structure of TCs, the organization of cloud clusters, and other characteristics. Therefore, satellite observations bring more critical perspectives and insights to researchers and more possibilities for understanding TC activities. Relevant studies have shown that the cloud structure presented by satellite images and the richness of meteorological data, such as temperature and humidity, can help estimate the intensity and scale of TCs [9–15]. In addition, each pixel of the infrared image from satellite observations corresponds to a bright temperature value, presenting the temperature distribution of the TC cloud, and at the same time reflecting the strength of the convective organization of the cloud [16–19].

In parallel with the development of observational techniques, TC intensity prediction models have been improved. Traditional forecasting techniques include numerical models and statistical models. The former simulates the behavior of the atmosphere by using kinetic and thermodynamic equations to reproduce the TC process and predict future changes in intensity. One of the more typical methods is the Hurricane Weather Research and Forecast system (HWRF) [20], which considers historical data that have a strong influence on the TC intensity changes and builds empirical models to predict the intensity changes (Climatology and Persistence model 5 day (CLIPER5)) [21]. Modern forecasting technology is based on deep learning methods. When building a network model, they are trained on a large amount of reanalysis data related to TC intensity changes and the relationship between TC intensity and these data is explored, such as considering the relative sea level temperature (SST), relative humidity (RH), vertical wind shear (VWS), etc. [22–25]. Recurrent neural networks have been applied in TC intensity prediction studies given their strong ability to capture temporal sequences. Kumar et al. [26] used Long Short-Term Memory(LSTM) to predict the intensity, location, and time of TC landfall by testing it in the Northern Indian Ocean. Chen et al. [27] extracted fusion features of satellite images and atmospheric environmental factors to predict future intensity changes using a deep learning integrated model. Na et al. [28] added artificial virtual data to the dataset and predicted the TC intensity and track using an echo state network. In addition, Pan et al. [29], Jiang et al. [30], Zhou et al. [31], Balaguru et al. [32], etc. constructed models based on Recurrent Neural Networks (RNNs) or their variants to achieve TC intensity prediction. Since convolutional neural networks (CNN) can be invariant to transformations such as rotation and cropping, Wang et al. [24], Ma et al. [33], Lee et al. [34], etc., have extracted the complex features of satellite images or reanalysis data based on CNN and established a link between the features and the target to achieve intensity prediction. Considering the above spatial and temporal features together, Convolutional Gated Recurrent (ConvGRU) has been proposed to be applied to the processing of image sequences [33,35,36].

Most of the above studies have achieved intensity prediction by constructing a link between reanalysis data and intensity changes. However, reanalysis data are used to provide a more accurate historical record of TCs by integrating the collected data again, processing the noise, missing values, etc., in the data, simulating and reconstructing the TCs, and finally carrying out data calibration and evaluation, thus providing a more accurate historical record of TCs. Relevant studies illustrate that the use of reanalysis data helps improve the accuracy of forecasting TC intensity, but in practical business applications, it is

difficult to meet the urgent need for the national forecasting of TC intensity because the reanalysis information is not available in real-time [35,37–39].

In summary, deep learning methods are important in TC intensity prediction research, but there are still some shortcomings. First, the non-real-time nature of the data. The reanalysis data are obtained from historical data and numerical model reconstruction, rather than capturing the evolution of TC in real-time, which has a non-real-time nature [40–42]. Second, the physical consistency is weak. Most of the current predictions of TC intensity focus on exploring the relationship between atmospheric reanalysis data and intensity, without considering the physical processes and mechanisms of TCs, and ignoring the effects of structural changes in the TCs themselves on intensity [33,43,44]. Finally, different information is treated uniformly. In TC intensity prediction studies, there are often multi-sources data or different information, but most of their features are usually simply merged and no measures are taken to differentiate between attention or learning information [34,45].

To address the above issues, we utilize deep learning to construct a model that deeply explores the relationship between multi-source data and intensity, aiming to improve the accuracy of TC intensity prediction. We use real-time observed satellite images to capture the evolution of TCs. The high spatial resolution of satellite images provides detailed cloud map features of the TC [46,47]. In addition, in order to fully learn the physical processes occurring during the TC intensity change, we consider the TC convective structure, the lag between intensity change and convective activity, and the symmetry of the convective organization of the TC cloud are characterized by the declination angle variance (DAV) while analyzing and predicting the intensity [48,49]. In addition to this, 3D CNN and 3D ConvGRU are used to capture spatial and temporal features, construct a model to extract the fusion features of satellite images and DAV, set up a double-branch structure due to the differences in the information contained in the satellite images and DAV, and set up AWL to automatically adjust to learn the differences between the double-branch information.

Details of the data and methodology are presented in Section 2, and Section 3 discusses the contributions of the modules. Section 4 evaluates the model performance through extensive experiments. Section 5 draws conclusions.

## 2. Data and Methods

### 2.1. Data

Predicting TC activity is more difficult than other weather phenomena because the formation and development of TCs involves complex interactions, including ocean–atmosphere interactions, interactions of TC convective activity, etc. [50,51]. In addition, TC is a large-scale weather system with rapid evolution and intensity changes influenced by a variety of factors. Therefore, this study introduces the basic parameters of TC monitoring (latitude, longitude, scale), satellite observation images, and computed DAV Hovmöller diagrams, which are closely related to the TC intensity, aiming to capture and learn the correlation between the above multi-source data and intensity. The northwest Pacific (NWP) region is a more active area where TC occurs or evolves, compared to other seas [52]. Therefore, this study is an experiment on NWP TC samples.

Basic parameters of TC monitoring and satellite observation data use the TCIR dataset provided by Chen et al. [53], which is widely used in the field of TC research [27,54,55]. This dataset integrates satellite observations from four channels from 2003 to 2017, including the TC center location (Lon, Lat), maximum sustained wind speed (Vmax), 35 kt maximum wind radius (R35), minimum sea level pressure (MSLP), detailed in Table 1. Infrared (IR1) shows the bright temperature of TC cloud tops, and cooler bright temperature indicates higher cloud tops, suggesting stronger convection. Water vapor (WV) shows the distribution of atmospheric water vapor content, and higher water vapor content is supportive of TC development. Passive microwave rain rate (PMW) provides precipitation, presenting the spatial distribution of precipitation. Visible light channel (VIS) shows the brightness and color of clouds, but it is not considered in this study because of its unstable observations

due to the influence of insolation and cloud obscuration. The spatial resolution is 0.07° for IR1, WV, and VIS, and 0.25° for PMW. To unify the spatial resolution of PMW and the other three channels, it was adjusted to 0.07° using linear interpolation, i.e., the distance between the two grids was 4 km. The images in the TCIR dataset were stored at a size of 201 × 201 and the center of the TC was located at the center of the image.

**Table 1.** Details of the TCIR dataset. IR1 provides the IR brightness temperature, WV provides the water vapor distribution, and PMW provides the precipitation.

| Channel | Source | Time Resolution | Wavelength | Monitoring Parameter |
|---------|--------|-----------------|------------|----------------------|
| **IR** | GridSat | 3 h | 11 μm | Lat, Lon, Vmax, MSLP, R35 |
| **WV** | GridSat | 3 h | 6.7 μm | |
| **PMW** | CMORPH | 3 h | 85 Ghz | |

The TCIR dataset is collected from two publicly available data sources, Gridded Satellite data (GridSat) and CPC MORPHing technique (CMORPH). GridSat is a satellite information product based on the gridded processing of satellite data from meteorological geostationary satellites. It provides a high-resolution cloud map showing information such as the cloud structure, cloud top temperature, and cloud top height of TCs, which is helpful to study the development trend of TC structure and intensity, and provides key parameters such as TC track, center location, and maximum sustained wind speed to predict the future track and intensity of TC. CMORPH is a precipitation estimation technique that combines observations from polar-orbiting and nonpolar-orbiting satellites to generate high-resolution precipitation estimates, which helps to analyze the precipitation characteristics of TCs and to understand the impact of water vapor on TC development. In this study, we first fuse satellite remote sensing data from multiple platforms to enhance the diversity of data, compensate for the limitations of multi-platform satellite remote sensing data, achieve complementary information, and improve the quality of data, which helps to provide more comprehensive and rich detailed information about TCs, and boosts the model to better extract and learn the features that are highly correlated with the changes in the intensity of TCs. In addition, we extracted storm names, storm IDs, timestamps, intensities, TC intensity *categories*'s names, latitude and longitude, central pressure, and other fields from the best track data provided by the Joint Typhoon Warning Center (JTWC). These data were then linked to the basic information provided by TCIR as supplementary data.

In addition, the best track provided by the JTWC is linked to the basic information provided by the TCIR as auxiliary information.

When the TC develops and strengthens from an unorganized structure, the cloud appears deeply convective and symmetric. Given the lag between the TC intensity and the evolution of its own convective structure, we extract the bright temperature value of each pixel point, i.e., the infrared bright temperature (IRBT), from the infrared channel of satellite observations. The angle $\theta$ made by the IRBT gradient vector at each pixel point with the line connecting that pixel point to the center was calculated, and the variance of the absolute value of $\theta$ was calculated for all pixel points as DAV. When the IRBT is more concentrated, the TC cloud is highly symmetric and the DAV is smaller. Therefore, the DAV characterizes the degree of symmetrization of the global organization of the TCs, and at the same time reflects the strength of the TCs. The Hovmöller diagram is a tool for visualizing and analyzing the variation of the potential with time and longitude [56]. In this study, Hovmöller diagrams are plotted to explore the variation of DAV with time and TC radius, and the dynamic evolution of TC convective structure is presented in a static way to achieve the downscaling of high-dimensional information, and the specific process is shown in Figure 1. In addition, due to the rotational invariance of the TC satellite image, we rotated and cropped the satellite image before inputting it into the model. The rotation angle was chosen randomly between −10° and 10°, while the specifics about cropping are described in detail in Section 2.3.

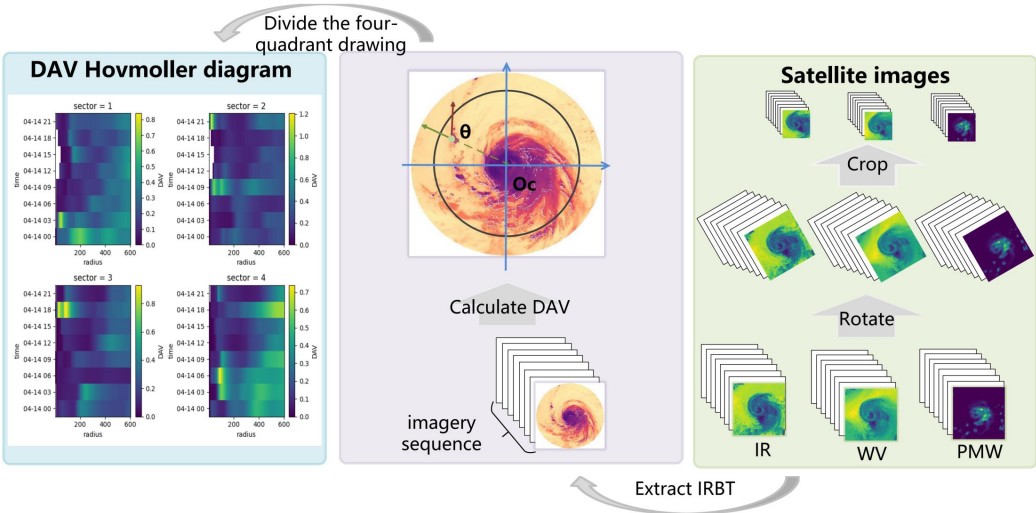

**Figure 1.** Three-channel satellite images (green background block) used in this study as well as the mapping of the DAV Hovmöller diagrams (purple and blue background block).

Since the overall shape of the TC is circular, in order to prevent the calculated DAV from being smoothed to affect the symmetry of the original TC, we divide it into four quadrants to calculate the DAV separately and plot the corresponding Hovmöller diagrams.

### 2.2. Model and Methods

As TC intensity changes undergo complex physical processes, in order to consider physical consistency, we introduce IRBT and construct the DAV Hovmöller diagram, from which we analyze the influence of structural changes within TC on the intensity. Therefore, we propose the model AWL-Net, which can effectively fuse multi-source data and analyze and learn the spatial and temporal correlation between the data and the targets, aiming to improve the accuracy of the model's prediction of the future 24 h TC intensity. The AWL-Net model and the functional modules are detailed in this section.

#### 2.2.1. Overview of AWL-Net Framework

Exploiting the long-term dependence between sequences is the key to predicting intensity in data-driven models. In order to accurately predict the future intensity changes of TCs, it is necessary to deeply explore the temporal sequences among data, and we propose the AWL-Net model, as shown in Figure 2. The model consists of a feature extraction module (FE) and a feature fusion module (FF), and AWL is set to assist the model analysis of data, aiming to explore the time dependence between data and target.

The grey area box in Figure 2 shows the workflow of the model. Firstly, the model receives multi-source data as inputs, namely multi-channel satellite observations and computed DAV Hovmöller diagrams. Set up a double-branch structure where each branch is used to process different types of data. Diverse information about TCs provided by multi-source data is analyzed by the FE module. Due to the differences between the information of the multi-source data, we consider embedding AWL in the double-branch structure to update the model parameters, so that it automatically adjusts and updates the appropriate weights according to the needs of the model, so that the model can better learn the relationship between the data and the target. The extracted features are then dimensionally matched and fused to obtain a more comprehensive and enriched feature representation that enhances the model's prediction of the target.

#### 2.2.2. Feature Extraction Module

The FE module can improve the understanding of the model, such as the green area block in Figure 2. Since the TC intensity has strong time dependence, the model should focus on the temporal features while capturing the spatial features of the image, so this

module is connected by multiple 3D ConvGRUs. The green area block of Figure 2 presents two parallel branches of ConvGRU. ConvGRUs inherit the advantage of convolution operation to deal with local spatial structure and features and have a good understanding of spatio-temporal data with local correlation, and at the same time inherit the GRU's capability of capturing the long-term temporal dependencies and has a strong understanding of temporal data with temporal correlations.

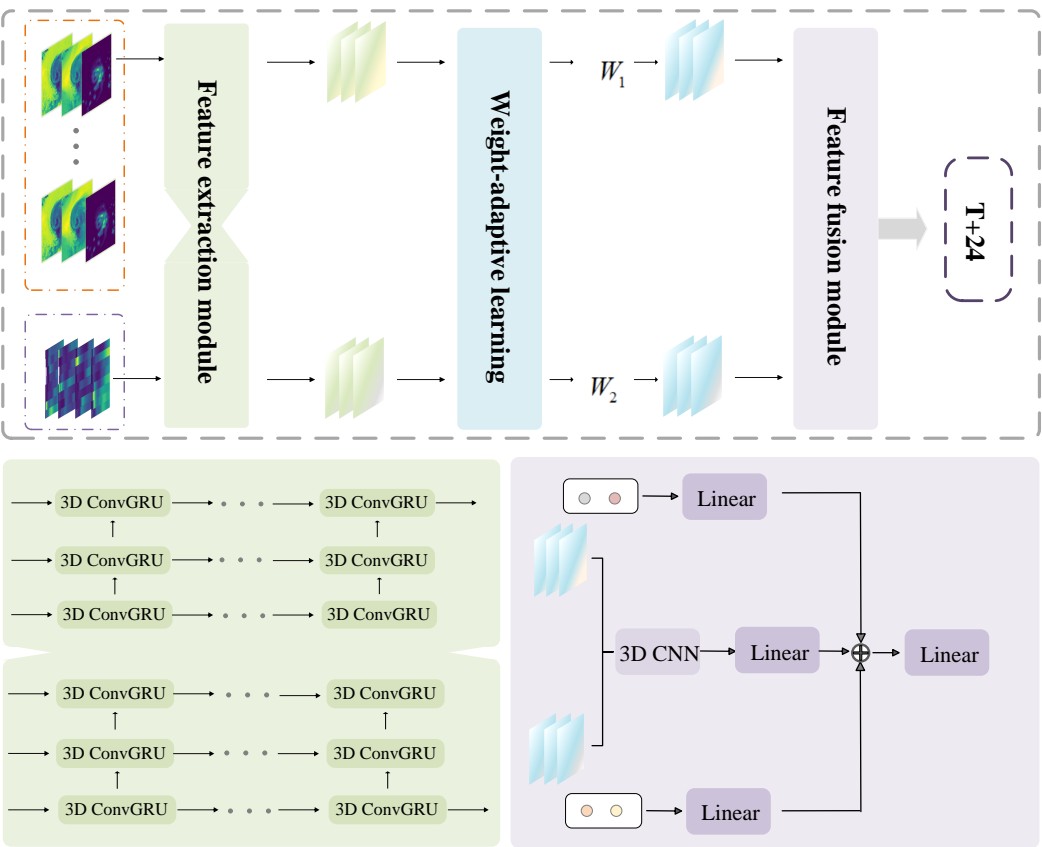

**Figure 2.** Schematic representation of the AWL-Net model framework. The grey area box is the model backbone, and the green and purple areas are the corresponding feature extraction and feature fusion in the backbone network, respectively.

ConvGRU can extract localized and hierarchical features compared to ConvLSTM and Transformer, which are widely used in time-series data processing. ConvGRU focuses only on locally correlated data, avoiding information interference from the surrounding data, and extracts hierarchical feature representations by controlling the flow of information through gating units. In addition, ConvGRU parameters are efficient. As the convolution operation enables parameter sharing, fewer parameters are required, making it easier to iterate, train and optimize when performing large-scale time-series data extraction and analysis. For the data used in this study, we use 3D ConvGRU to construct a double-branch feature extraction to explore the information embedded in IR, WV, and PMW, as well as the TC cloud symmetry presented by the static Hovmöller diagram, respectively.

### 2.2.3. Adaptive Weight Learning

The attention mechanism assigns different weights according to the level of regional information and relevance, so that the model has a difference in focus when learning. AWL analyzes the correlation between different input data and the target, assigning different weights to the feature extraction branches to accommodate the relationship between different data and the target.

Due to the differences in TC information provided by satellite observations and DAV Hovmöller diagram, AWL is used to overcome the problem of single double-branch parameters and difficult to adjust. In the double-branch learning process, the weight parameters are adaptively adjusted by different data distribution characteristics and different differences with labels, so that the model better adapts to the complexity of the input data and timely adjusts the learning process of different branches. The adaptive learning process of weights can be defined as follows:

$$W_i = \frac{e^{w_i}}{\sum_j e^{w_i}} (i = 1, 2; j = 1, 2) \tag{1}$$

where $w_i$ denotes the initialized weight coefficients and $W_i$ denotes the weight coefficients of a particular branch.

### 2.2.4. Feature Fusion Module

Multi-source data provides different reference information, effective extraction also requires efficient fusion, and simple connection can easily cause the fuzzy processing of information. The FF module can enhance the expressive ability of the model, and the design of this study is jointly constructed by 3D CNN and Fully Connected Layer (FC), as shown in the purple area block in Figure 2. The two blue gradients in the figure are different features extracted from the double-branch under different weights, as follows. The temporal and spatial features are firstly fused by 3D CNN processing, and then stretched to one-dimensional by FC.

$$F = Conv((W_1 \times F_1) \otimes (W_2 \times F_2)) \tag{2}$$

where $W_1$, $W_2$ denotes the double-branch weights, $F_1$, $F_2$ denotes the features extracted from both branches and $\otimes$ denotes the convolution operation.

Considering that latitude and longitude provide information on TC location, the scale size is a characterization of the strength of convective activity. Meanwhile, strong convective activity requires a larger scale to be maintained, and when the TC is in the decaying phase, the scale decreases due to the weakening of the intensity. Therefore, we introduce the Lat, Lon, scale (R35) and 24 h historical intensity time series as auxiliary information for intensity prediction, presenting the evolution of convection and intensity, which corresponds to the four points of grey, pink, orange, and yellow in the purple area block of Figure 2. Firstly, the one-dimensional time series data are processed by FC to capture the temporal dependence between values. Secondly, the acquired one-dimensional information is fused with the information extracted from the high-dimensional features, and finally the prediction results are output by FC processing.

### 2.3. Data Processing

The data from TCIR from 2003 to 2017 are firstly divided, and the specific division method as well as the number of TCs and the number of data are shown in Table 2. In order to prevent the interference of the edge information in the data as well as the overfitting of the model training data and other problems, we preprocessed the data. Based on the rotational invariance of the TC images, we rotated and cropped the satellite data. Taking into account the cloud structure and morphology of the TCs and the range of image sizes that can be processed by the model, we set up two sets of experiments to further confirm the image sizes, as shown in Tables 3 and 4. Raw image size is 201 × 201. Table 3 examines the effect of different sizes of satellite images on the experiment, selecting sizes with larger interval and initially selecting the size. Table 4 is the discrimination based on Table 3, based on the value of the size derived from Table 3, around which the interval is reduced to further determine the optimal size. The best results are shown in **bold**. From Table 3, it can be seen that the RMSE is decreasing by the gradual cropping from the original image to the 101 × 101 image, while it increases when going from 101 × 101 to 51 × 51. Therefore,

in Table 4 we will focus on the change of RMSE at the stage of image cropping from $101 \times 101$ to $51 \times 51$. From Table 4, the satellite image with $95 \times 95$ pixels has the smallest RMSE, which means that the $95 \times 95$ pixels image is more appropriate for the TC intensity prediction study and for the dataset we are currently using. It is also worth noting that the above experiments did not change other elements except for the size of the satellite images. In addition, null or damaged samples present in the data are set to NaN. Since this study addresses the prediction of TC intensity, the satellite images were sorted chronologically and the historical 24 h data were integrated into a time series.

Satellite imageries and the generated DAV Hovmöller diagram are not immune to the presence of singular data, which are normalized separately in order to prevent the effect of singular data and data at different scales as follows.

$$x_{norm} = \frac{x_i - \bar{x}}{q}, q = \sqrt{\frac{1}{n} \sum (x_i - \bar{x})^2} \tag{3}$$

where $x_i$ and $\bar{x}$ are the input data and the mean of the input data, respectively, and $x_{norm}$ and $q$ are the normalized value and variance.

**Table 2.** The division of the dataset.

| Set | Temporal Interval | Number of TC | Number of Data | Ratio (%) |
|---|---|---|---|---|
| Training set | 2003–2013 | 265 | 12,562 | 67.7 |
| Validation set | 2003–2013 | 33 | 1308 | 7.1 |
| Testing set | 2014–2017 | 115 | 4684 | 25.2 |
| Total | 2003–2017 | 413 | 18,554 | 100.0 |

**Table 3.** RMSE of satellite images of different sizes (large interval).

| Size | Channels | RMSE(kt) |
|---|---|---|
| $201 \times 201$ | 3 | 12.91 |
| $151 \times 151$ | 3 | 11.53 |
| $101 \times 101$ | 3 | **11.03** |
| $51 \times 51$ | 3 | 11.74 |

**Table 4.** RMSE of satellite images of different sizes (small interval).

| Size | Channels | RMSE(kt) |
|---|---|---|
| $115 \times 115$ | 3 | 11.16 |
| $105 \times 105$ | 3 | 10.85 |
| $95 \times 95$ | 3 | **10.62** |
| $85 \times 85$ | 3 | 10.73 |
| $75 \times 75$ | 3 | 11.01 |
| $65 \times 65$ | 3 | 11.32 |
| $55 \times 55$ | 3 | 11.72 |

The AWL-Net model is built based on PyTorch 3.8, the experimental configuration is GeForce RTX 2080ti 11 GB GPU, CPU is Inter Core i9-9900K. In the early stage of model building, we conducted a large number of experiments to evaluate the effect of different parameter combinations on the model performance. By evaluating the model performance, the Adam optimizer is selected for optimization, the learning rate is set to 0.0005, and the learning rate is adjusted regularly using the scheduler to prevent gradient explosion or overfitting problems. In addition, based on the limitation of computer resources and the analysis of experimental results, the batch size is set to 16.

*2.4. Evaluation Metrics*

To evaluate the prediction performance of our proposed model, the following evaluation metrics are used: mean absolute error (MAE), mean absolute percentage of error (MAPE), root-mean-square error (RMSE), and r-squared error ($R^2$). MAE is used to measure the degree of bias in the predicted values. MAPE is the percentage of error between the predicted and true values, reflecting the relative degree of bias in the predicted values. RMSE is a reflection of the degree of accuracy in the predicted values. $R^2$ measures the degree of fit of the model predictions to the true values. These indicators are calculated as follows:

$$RMSE = \sqrt{\frac{1}{N} \sum_{i=1}^{N} (y_i - \hat{y}_i)^2} \tag{4}$$

$$MAPE = \frac{1}{N} \sum_{i=1}^{N} \left| \frac{y_i - \hat{y}_i}{y_i} \right| \tag{5}$$

$$MAE = \frac{1}{N} \sum_{i=1}^{N} |y_i - \hat{y}_i| \tag{6}$$

$$R^2 = 1 - \frac{\sum_i (\hat{y}_i - y_i)^2}{\sum_i (\overline{y_i} - y_i)^2} \tag{7}$$

where $y_i$ and $\hat{y}_i$ denote the actual and predicted values at a given moment, $N$ is the number of time points to be predicted, and $\overline{y_i}$ is the average of the true intensity values.

## 3. Model Evaluation and Discussion

*3.1. Evaluation on Module Contribution*

In previous TC intensity prediction studies, the relationship between environmental variables or satellite observations and intensity is only singularly explored, and the influence of the evolution of TC convective structure on intensity changes is seldom considered. For multi-source data are simply connected to achieve feature fusion, resulting in the fuzzy processing of features. In contrast, AWL-Net focuses on the information provided by satellite observations while paying attention to the evolution of the TC organizational structure, presenting the symmetry of the clouds with the DAV Hovmöller diagram, so as to analyze the future trend of intensity change. And the FE and FF modules are designed for time series data to process the data efficiently, aiming to improve the model performance. The contribution of the module design in the model and different channels will be illustrated by ablation experiments in the following, as shown in Table 5. The △ denotes the improvement compared to the benchmark, with the optimal results shown in **bold** and sub-optimal results underlined. Note that the optimal and sub-optimal results for the data inputs and the optimal results for the modules are shown in Table 5.

The first is a discussion of the input data. We examine the effects of single-channel, two-channel, and three-channel combinations, as well as the DAV Hovmöller diagram, on the experiments. A base model is constructed from 3D ConvGRU to analyze different inputs' contributions to improving intensity prediction accuracy. As shown in Table 5, the introduction of a three-channel combination (Multi-chan.) has the best effect on improving the model performance. This is due to the fact that the three-channel is rich in information, which can describe the TC more comprehensively by presenting the bright temperature of the cloud tops while reflecting the distribution of water vapor and microwave radiation. The DAV Hovmöller diagram is the following best input in addition to Multi-chan., suggesting that DAV can be useful in exploring the mechanism of structural changes within TCs and sudden changes in intensity over a short period.

The performance of the FE module is discussed below. The FE(GRU) in Figure 3 denotes that the FE module is connected by 3D ConvGRUs to extract features of three-channel fusion and cloud organization structures by the double-branch, respectively, while FE(CNN) denotes that the 3D ConvGRUs in the FE module are replaced by 3D CNNs. Ob-

viously, the use of the 3D ConvGRU can model the spatio-temporal dependence effectively. Because it is a combination of CNN and GRU, extracting local features can recursively process the data while improving the data utilization.

**Table 5.** Different inputs and network module contributions.

| Input | Module | MAE(kt) | △ (%) | RMSE(kt) | △ (%) |
|-------|--------|---------|-------|----------|-------|
| IR | 3D ConvGRU | 10.69 | - | 14.14 | - |
| WV | 3D ConvGRU | 10.71 | −0.18 | 14.17 | −0.21 |
| PMW | 3D ConvGRU | 11.64 | −8.89 | 15.06 | −6.54 |
| IR + PMW | 3D ConvGRU | 10.56 | 1.22 | 13.80 | 2.40 |
| WV + PMW | 3D ConvGRU | 10.57 | 1.12 | 13.89 | 1.77 |
| IR + WV + PMW (Multi-chan.) | 3D ConvGRU | **10.11** | 5.43 | **13.43** | 5.02 |
| DAV | 3D ConvGRU | 10.48 | 1.96 | 13.65 | 3.47 |
| Multi-chan. + DAV | FE(CNN) | 9.53 | 10.85 | 12.63 | 10.68 |
| Multi-chan. + DAV | FE(GRU) | 8.64 | 19.18 | 11.39 | 19.45 |
| Multi-chan. + DAV | FF(GRU) | 8.43 | 21.14 | 11.23 | 20.58 |
| Multi-chan. + DAV | FF(CNN) | 8.35 | 21.89 | 11.03 | 21.99 |
| Multi-chan. + DAV | FF(CNN)+AWL | 8.13 | 23.95 | 10.71 | 24.26 |
| Multi-chan. + DAV + 1D Data | AWL-Net | **8.07** | 24.51 | **10.62** | 24.89 |

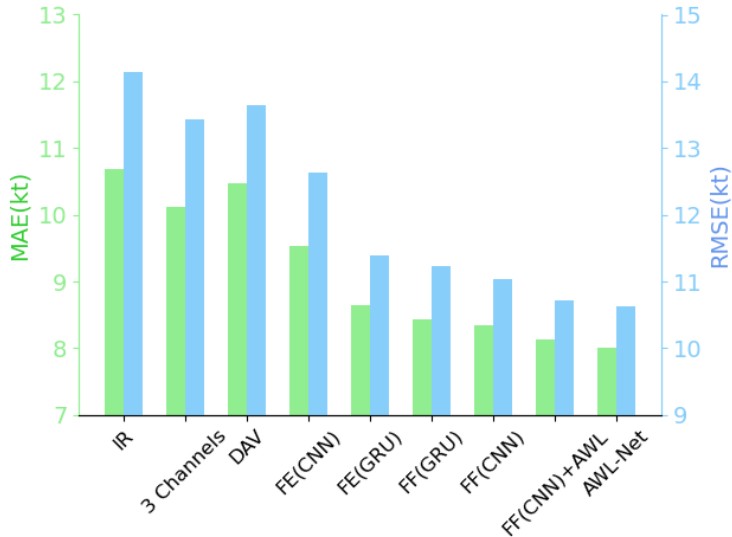

**Figure 3.** Evaluation on MAE and RMSE of the ablation experiments.

In Figure 3, FF(GRU) denotes that the FF module consists of 3D ConvGRUs, and FF(CNN) denotes that 3D CNNs are used instead of 3D ConvGRUs in the FF module. By comparing the prediction errors between them, it can be seen that the FF module we designed can fuse the data efficiently. Although it is illustrated in the FE module that 3D ConvGRU is highly capable of modeling spatio-temporal dependencies, different networks should be selected for different tasks. When fusing the data, we are more concerned with the integration of the extracted static features rather than the temporal order, and we do not need to rely on the temporal information for the cyclic operation, so the 3D CNN will outperform the 3D ConvGRU when feature aggregation is performed.

For AWL, FF(CNN) is compared with FF(CNN) + AWL in Figure 3, and it is clear that the model with AWL setup predicts smaller MAE and RMSE. This is because the data analyzed in the double-branch analysis are very different and describe the TC in different ways. Therefore, allowing the model to learn by automatically assigning weights as needed greatly improves the learning process of the two different branches and makes it easy to construct the relationship between the two types of data and the target.

AWL-Net is the complete model architecture we propose, and the inputs include the three-channel satellite imageries, the DAV Hovmöller diagram mentioned above, in addition to one-dimensional TC descriptive information (Lat, Lon, R35, history intensity), denoted as 1D-Data in Table 5. AWL-Net has fewer prediction errors than FF(CNN) + AWL. It can be demonstrated that these one-dimensional data are complementary to the model for predicting TC intensity. The center of the TC and R35 are a visual description of the TC structure. Thus, they provide a more comprehensive description of the historical state of the TC. As shown in Figure 4, there is a strong correlation between them and TC intensity. There is a highly positive correlation between R35 and TC intensity, and a highly negative correlation between the DAV Hovmöller diagram and TC intensity. It is further found that the difference in correlation between R35, DAV, and intensity in 2015 and 2017 may be due to the effect of different meteorological conditions such as SST, barometric pressure, humidity, etc., and also the effect of TC activities including tracks and number. In conclusion, the high correlation between them indicates that it is feasible for us to achieve intensity prediction by analyzing and extracting features from these related data. By establishing the relationship between scale (R35) and intensity and between cloud symmetry change (DAV) and intensity, the prediction model is constructed to allow the model to further understand the mechanism of TC formation and development, and to provide a scientific basis for predicting the TC 24 h intensity with improved accuracy.

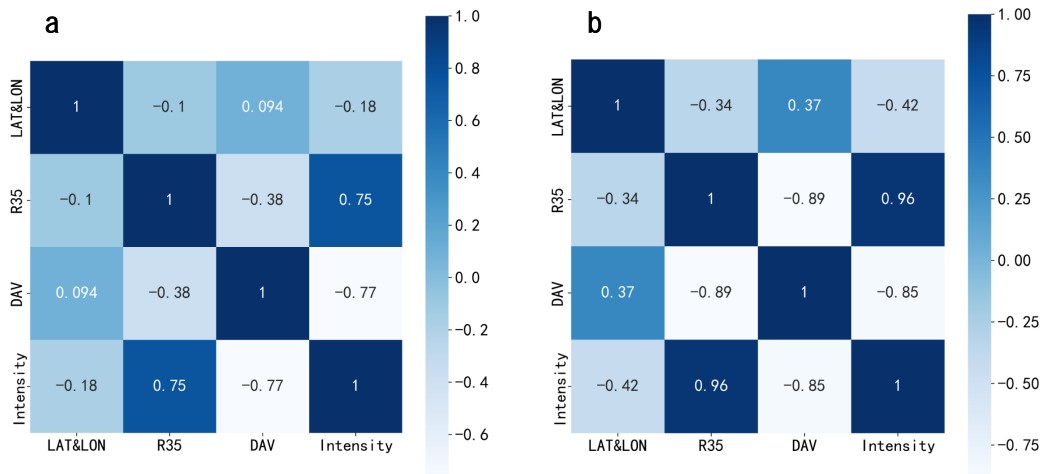

**Figure 4.** (**a**) Matrix of correlation coefficients for basic information on TC intensity values for 2015. (**b**) Matrix of correlation coefficients for basic information on TC intensity values for 2017.

## 3.2. Evaluation on Module Performance

To validate the performance of our model, this is evaluated using time and space complexity. Time complexity is measured using FLoating-point OPerations (FLOPs), which indicates the order of magnitude of time required by the model in solving the problem. If the time complexity is too high, it will result in a significant amount of time required for model training and prediction. Therefore, reducing this complexity can improve the computational efficiency of the model. Space complexity is measured using the amount of access memory (MAC), which indicates the order of magnitude of memory space required in solving the problem, and the general memory consumption includes model parameters, feature maps, etc. Higher spatial complexity implies higher memory consumption of the model; therefore, reducing the spatial complexity improves the memory utilization of the model. The time and space complexity of our model is shown in Table 6; ConvGRU1 is the feature extraction of multi-channel satellite imageries, and ConvGRU2 is the feature extraction of DAV Hovmöller diagrams, which contains only one forward propagation computation, and if backpropagation is included, it is 5.52 GFLOPs, and MAC is 133.2 MB.

**Table 6.** Time and space complexity of AWL-Net.

| Layer Type | Kernel Size | Kernel Mem. | Output Mem. | FLOPs |
|---|---|---|---|---|
| ConvGRU1_1 | (1,3,3,64,64) | 37,632 | 230,400 | 135,475,200 |
| ConvGRU1_2 | (2,7,7,64,128) | 221,184 | 230,400 | 796,262,400 |
| ConvGRU1_3 | (1,3,3,128,128) | 1,605,632 | 13,824 | 173,408,256 |
| ConvGRU1_4 | (2,7,7,4,64) | 884,736 | 13,824 | 95,551,488 |
| ConvGRU2_1 | (1,3,3,64,64) | 50,176 | 288,000 | 225,792,000 |
| ConvGRU2_2 | (2,7,7,64,128) | 221,184 | 288,000 | 995,328,000 |
| ConvGRU2_3 | (1,3,3,128,128) | 1,605,632 | 9216 | 115,605,504 |
| ConvGRU2_4 | (3,3,3,128,256) | 884,736 | 9216 | 63,700,992 |
| 3D CNN | - | 884,736 | 2048 | 159,252,480 |
| 3 × FC1 | - | 1,212,416 | 128 | 1,212,416 |
| 2 × FC2 | - | 2304 | 64 | 2304 |
| 2 × FC3 | - | 2304 | 64 | 2304 |
| 2 × FC4 | - | 32,896 | 1 | 32,896 |
| - | - | - | - | - |
| - | - | Kernel Mem. (Total) | Output Mem. (Total) | FLOPS (Total) |
| Summary | - | 7,645,568 | 1,085,185 | 2,761,626,240 |
| In Units | | 29.16 MB | 4.14 MB | 2.76 GFLOPs |

One area of interest to meteorologists as well as researchers is the 24 h TC intensity prediction, the accuracy of which is crucial for the timely issuance of warning signals and the adoption of precautionary measures. In recent years, researchers have been working on improving TC intensity prediction models to enhance prediction accuracy. To assess the validity of our model, it will be compared with other methods. As more studies are now predicting the intensity of TCs based on reanalysis data, they are introduced here for informational purposes only due to their different input data. The model in Hu et al. [49] mentioned above was re-implemented and additionally the TC prediction record of the official agency was used as a baseline as shown in Table 7, with the best predicted results indicated by **boldface** characters.

DAV-IR was proposed by Hu et al. to predict the 24 h TC intensity based on DAV using infrared images. DAV is a description of the symmetry of the TC organization, and the IR channel raises the infrared observation of the bright temperature. Obviously its prediction results are less effective compared to our fusion data, and our model AWL-Net achieves the minimal error with an MAE of 8.07 kt and an RMSE of 10.62 kt. By comparing the MAPE, we find that AWL-Net's is 12.8%, which is 6.52% lower than that of DAV-IR, which indicates that the relative degree of bias in the predicted values of our model is less, and the prediction is more accurate. Analyzed in terms of $R^2$, it has a better fit. Our three-channel satellite image is a presentation of TC in the vertical direction, which contains more comprehensive and rich information. To verify our model's performance, we can compare it with official agencies' tropical cyclone records. China Meteorological Administration (CMA) is the meteorological agency responsible for monitoring and forecasting weather changes in the region surrounding China's indigenous machines, combining observational data with numerical models to predict TC intensity, which is dedicated to safeguarding. The JTWC is a U.S. Department of Defense agency that monitors a region that includes the Northwest Pacific, North Pacific, South Pacific, Indian Ocean, and parts of the Atlantic Ocean. This region covers Asia, Oceania and parts of the Americas. JTWC has extensive experience in TC track and intensity prediction using real-time data from satellite observations and aircraft reconnaissance data to provide important information for early warnings in the relevant areas. The National Hurricane Center (NHC), an agency of the National Oceanic and Atmospheric Administration (NOAA), monitors and forecasts TC activity in the Atlantic and Northeast Pacific. It utilizes aircraft reconnaissance to provide real-time data and meteorological radar to monitor TC precipitation and storm structure. The NHC also employs multiple numerical models for TC intensity prediction. Our model also

outperforms the TC records of official agencies by 6.5%, 2.2%, and 2.8% relative to the CMA, JTWC, and NHC methods, respectively, thus proving the validity of the proposed model.

**Table 7.** Evaluation of the different models (MAE:kt).

| Method | Model | MAE | MAPE (%) | RMSE | $R^2$ | FLOPs (FLOPs) | MAC (Byte) | CD (FLOPs/Byte) |
|--------|-------|-----|----------|------|-------|---------------|------------|------------------|
| Deep Learning | Hybrid CNN-LSTM [22] | 9.96 | - | 12.58 | - | 7.36 G | 275 M | 26.76 |
| | TC 3D CNN [24] | 11.86 | - | 15.98 | - | 2 G | 118 M | 16.95 |
| | TC_Pred [35] | 8.19 | - | 10.79 | - | - | - | - |
| | DAV-IR [49] | 11.81 | 19.32 | 14.82 | 0.76 | - | - | - |
| | AWL-Net | **8.07** | 12.80 | **10.62** | 0.80 | 5.52 G | 133.2 M | 41.44 |
| Official Agencies | CMA | 8.63 | - | - | - | - | - | - |
| | JTWC | 8.25 | - | - | - | - | - | - |
| | NHC | 8.30 | - | - | - | - | - | - |

In addition to the prediction performance, we compare the spatio-temporal complexity to evaluate our model more comprehensively. From Table 7, we can see that the FLOPs and MAC of our model are the smallest except TC 3D CNN, which is since the spatio-temporal complexity is related to factors such as the model's convolutional kernel size, the number of channels, and the size of the output feature maps. The TC 3D CNN, on the other hand, only uses three-dimensional convolutions with a smaller number of channels and fewer convolutional layers. As a result, this model has the smallest FLOPs and MAC. Although our model is not optimal in terms of FLOPs and MAC, the computational density (CD) can reflect the execution speed and learning ability of the model as a whole. Therefore, considering the prediction performance and complexity, AWL-Net performs better.

## 4. Analysis and Discussion of Results

### 4.1. Study on TC Intensity Prediction

This subsection discusses the model's effectiveness in predicting TC 24 h intensity values. The fusion features of satellite imageries and the Hovmöller diagram are mined by 3D ConvGRU and combined with the TC location and scale to learn the intensity change rule and analyze the future intensity change trend. As shown in Figure 5a, the black dashed line is the ideal fit line, the overall trend of the predicted values and the real values are basically consistent, the predicted values are evenly scattered on both sides of the fitting line, and RMSE is 10.62 kt, which proves the high accuracy of the AWL-Net. However, there are obviously low predictions of strong TCs in individual samples. Figure 5b shows the scatter density distribution of predicted and true values, with different colors in the figure representing the corresponding densities, MAPE is 12.8%. It can be found that the data samples of weak TCs are significantly higher than those of strong TCs, which explains why the above problem exists. Fewer samples are difficult to train and learn, and the prediction results are slightly inferior to the TC strength of the multisample.

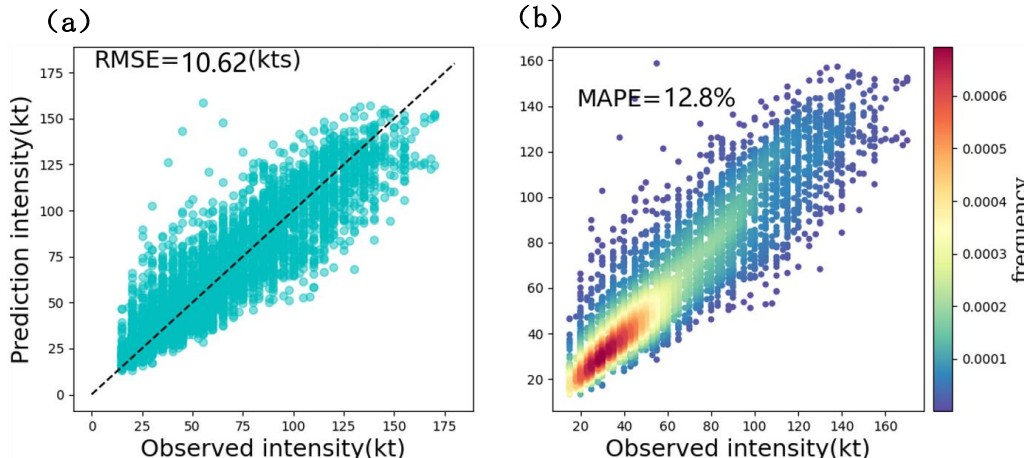

**Figure 5.** Evaluation on RMSE and MAPE of TC intensity prediction. (**a**) Predicted scatter plot. (**b**) Predicted scatter density plot.

*4.2. Study on Different Intensity Changes*

Since the TC intensity change is extremely unstable, there are abrupt changes, and more severe cases can occur with rapid intensification (RI) or rapid weakening (RW) (the maximum sustained winds are usually considered to have intensified or weakened by at least 30 kt in 24 h). Bai et al. introduced multi-domain knowledge to predict RI based on deep learning, and Zhou et al. constructed a TC RI identification model based on ResNet and LSTM to make predictions [57,58]. This subsection analyzes the change in intensity of TC over 24 h and compares the real and predicted changes in intensity over 24 h of TC, which will help to analyze the prediction bias of the model in a given case. These differences provide us with information about the limitations of the model and directions for improvement. As in Figure 6a, the number of TC samples with different intensity variations over 24 h in the test set is counted, and it can be seen from the figure that the intensity variations are more concentrated within 25 kt, and there are very few TCs whose intensity variations will be more than 75 kt. Figure 6b,c are the deviations reflecting the true and predicted intensity changes, and there is a large deviation in the model's prediction when the intensity increase value is greater than 55 kt. And the model predicts more accurate changes for TCs experiencing rapid weakening. This may be due to the fact that TCs experiencing RW show a distinct characteristic change due to some key environmental conditions and dynamical processes that are weakening, and our model successfully captures this distinctive feature. RI, on the other hand, is difficult to capture its characteristics or difficult to analyze because it is affected by more complex dynamics and environments, so it results in less predicted changes for samples with increasing intensity.

Through studying the predicted and real values of intensity changes within 24 h, it is clear that the current model predicts the intensification or weakening of intensity better, which means that the model performs well for the trend (positive or negative) of intensity changes within 24 h, and it is almost impossible to have a situation in which the TC is enhanced (weakened) while the model predicts it to be weakened (enhanced). It follows that one of the essentials affecting the performance of our model in predicting the intensity of TCs over the future 24 h is the inaccuracy of the model's prediction of the experienced RI. When the TC experiences large intensity changes, the corresponding intensity changes predicted by the model are smaller in magnitude, which leads to a larger gap between the model-predicted future intensity values and the true values. Therefore, the improvement of our research can focus on learning and analyzing strong TCs. The data enhancement techniques are used to expand the strong TC dataset, and richer data samples are constructed by rotating, scaling, and adding noise to remote sensing satellite images to enhance the learning ability of the model in the face of strong TCs. In addition, it might be possible to add an attention mechanism module to target training and learning of strong

TC cloud features and structural features, to improve the model's ability to adapt to strong TCs. This may also be a measure taken by our future work on predicting TC intensity.

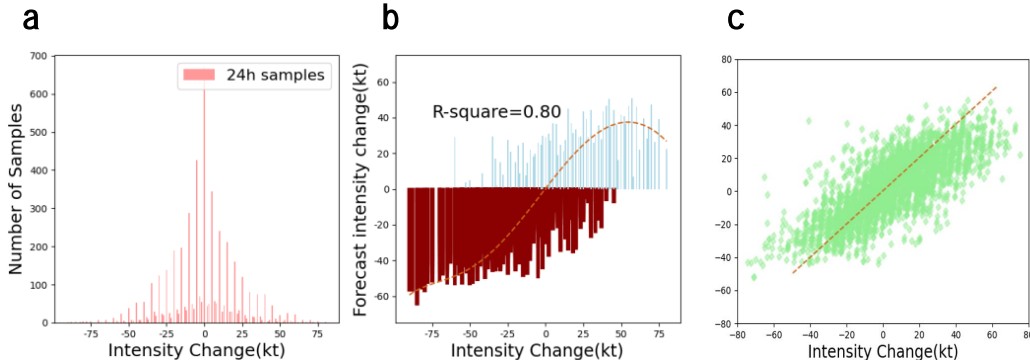

**Figure 6.** Prediction of 24 h intensity changes. (**a**) Sample number. (**b**) Predicted versus true 24 h intensity change. (**c**) Scatter plot of predicted versus true values for 24 h intensity changes.

### 4.3. Study on Different TC Categories

The discussion above has made it clear that there are differences in the sample sizes for different intensity variations, and that intensity variations over 24 h are more concentrated around 0 kt, so the intensity variations are smaller and the model is easier to learn and predict. However, the prediction error increases when the TC experiences a large RI, the possible reasons for this are also analyzed above, which is an area for subsequent research and exploration. To further clarify the causes of strong TCs but low predictions, we discuss the classification of TC intensities. In this subsection, the intensities are categorized according to the category classification criteria of JTWC as tropical depression (TD), tropical storm (TS), typhoon (TY), super typhoon (Super TY), as shown in Table 8, which also lists the intensity classification criteria for intensities in km/h and kt.

**Table 8.** Classification of TC intensity based on JTWC.

| Category's Names | TC Intensity (km/h) | TC Intensity (kt) |
|:---:|:---:|:---:|
| TD | 0–61 | 0–33 |
| TS | 63–117 | 34–63 |
| TY | 119–239 | 64–129 |
| Super TY | ≥241 | ≥130 |

Figure 7a counts the number of samples for the four TC categories, with TD, TS, and TY having higher numbers, all exceeding 1000, and Super TY having the lowest number of samples, accounting for only about 11.8% of the number of TS samples. Figure 7b is a box plot of the model's predictive performance for different TC categories. In the figure, the dark blue solid line indicates the median, which is the center of the data, and the pink dashed line indicates the mean of the data. It is clear that the model's prediction performance for TD, TS, and TY is better than that of Super TY, which is partly due to the imbalance in the distribution of the data. Super TY is a TC intensity value greater than or equal to 130 kt, and high-intensity storms are relatively rare in the TC historical record. Deep learning models need to learn features through a large number of samples in training, while the unbalanced distribution of data affects the learning process of the model. More attention is paid to TD, TS and TY with larger sample sizes, and more samples are accessible to analyze and learn. In addition to differences in the number of data samples, TD, TS, and TY have more stable external environmental conditions relative to Super TY, e.g., demand for ocean surface temperature and water vapor, and the dynamical processes are simpler, and the models are easy to analyze and capture their spatiotemporal characteristics. In contrast, Super TY undergoes complex dynamical and thermal processes, with extremes, and its characteristics are less regular and difficult to analyze and learn.

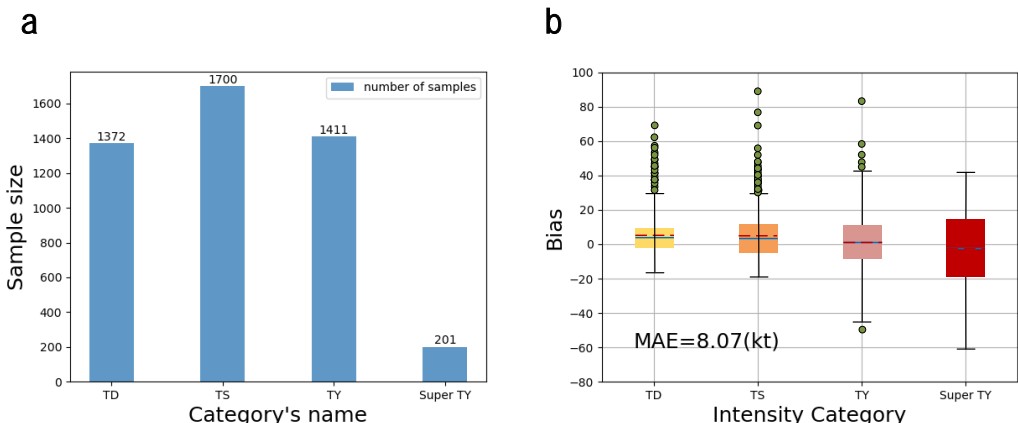

**Figure 7.** TC category sample and category prediction. (**a**) Number of TC category samples. (**b**) Deviation in the TC category prediction.

The categorical prediction of the intensity of TCs is the result of a refined assessment of the predictive performance of the model. TCs of different intensity classes have different characteristics and prediction difficulties, and the analysis of the categorical predictions clearly shows the strengths and weaknesses of the model as well as the difficulties in predicting future intensities. In other words, another essence that affects the performance of our model in predicting the intensity of TCs for the future 24 h is the inaccuracy of the model in predicting strong TCs. Therefore, improving the prediction capability of strong TCs is the key to improving the prediction accuracy of the model. By categorically discussing prediction performance, strategies can be developed according to the characteristics of different intensity levels. Prediction methods are simplified for weaker TCs, and data are expanded and complex prediction models are applied for stronger TCs.

The classification prediction of TC intensity is also a way to prove the generalization performance of the model. Our training does not classify the TCs according to their intensity, so the resulting model is not trained for a particular class or classes of intensity class TC samples. Only classify the TC samples from the test set at the test and input them into the model to obtain the prediction bias of different categories. The model has a good prediction performance for TCs of different intensity classes; therefore, our model has a strong generalization ability.

*4.4. Case Study*

To verify the generalizability and validity of the model, we will discuss the prediction of individual cases in this section. Individual case analysis allows an in-depth study of the characteristics and evolution of a specific TC. By inputting individual cases into the model, the performance of the model on specific events is evaluated, looking for limitations in the model's predictions. We have selected four representative examples of TCs, each with the following characteristics. TC HATO, No. 15 of 2017, which reached a maximum intensity of 90 kt, HATO rapidly progressed from initial TD to TY, experiencing a more pronounced intensification phase during its development and weakening immediately after landfall, as shown in Figure 8a. TC SANVU, No. 17 of 2017, both reached a maximum intensity of 90 kt relative to HATO, as shown in Figure 8b, but the intensification and weakening process of SANVU was more rapid, which means that it experienced a more significant increase and weakening of intensity in a shorter period of time. TC DOKSURI, No. 21 of 2017, which had a smoother and slower rate of growth, reached a peak intensity of 95 kt before landfall, and then weakened gradually as shown in Figure 8c. TC LAN No. 25 of 2017, which has a long life cycle as shown in Figure 8d, initially formed in the western Pacific Ocean and rapidly intensified to a TY of 93 kt in the initial phase, and then gradually moved northward to absorb heat and water vapor from the surrounding area, and once again increased in intensity to a peak intensity of 131 kt, becoming a Super TY. There is then

a small phase of a decline in LAN intensity, possibly due to changes in ocean conditions and topographic interactions, but it may once again reach peak intensity 135 kt due to re-entry into a suitable environment, and finally tapers off in areas close to the coastline, but it is worth noting that it is still a TS.

Figure 8 presents an overall presentation of the predicted intensity changes throughout the life cycle of the four TCs using the model AWL-Net; at the same time, the track information of the four TCs is provided to assist in analyzing the intensity changes through the changes in the TC tracks. The left graph of each set of plots shows the model predictions, with the horizontal axis showing the time from generation, development to extinction of TCs, and the vertical axis showing the intensity of TCs. The real intensity values of TCs (orange solid line) and the predicted intensity values (green solid line) are represented in different colors and line shapes, which makes them clearer in the figure. The right graph of each set of plots shows the track information of the TC, with horizontal and vertical coordinates corresponding to longitude and latitude, respectively, and the change in intensity is reflected in the change in color. As shown in Figure 8a,b, both TCs develop rapidly to TY and reach the peak, followed by a rapid weakening from TY, and the trend of intensity change is symmetric for the peak. However, SANVU has better intensification and remains a TS as SANVU dissipates, while HATO is a TD. Both HATO and SANVU are well-predicted by our model and can capture the peaks. Moreover, the models are not confounded by the different intensity ranges of the dissipation of the two. As shown in Figure 8c, the model fits the basic trend of DOKSURI's prediction better, and the model always predicts accurately when it experiences intensification or weakening. It shows that the model can more accurately capture the evolutionary characteristics of the TC for stable developmental changes, which is easier to learn. However, for the LAN, which has a faster intensity change, when its intensity exceeds 100 kt and reaches 128 kt, it suddenly turns from continued intensification to flatness, and the model does not capture this change well, and still predicts it as continued intensification, as shown by the red box in Figure 8d. This particular situation deserves our attention and discussion.

For the special case mentioned above, i.e., marked by the red box in Figure 8d, this may be due to insufficient data samples, and if this continued intensification turned to a flat change is missing in the training data, the model will have a hard time to predict accurately in feature extraction and learning without sufficient samples. In addition, the sudden change in TC intensity is itself a phenomenon involving complex physical processes, and there are more factors affecting the mutation, such as eyewall replacement, vertical wind shear, and the interaction between TC and land. Therefore, some relevant influencing factors can be considered to be added in the subsequent work to better capture the sudden change in TC intensity. Beyond that, it is hard to predict what the peaks of intensity will be. Although the sample size of strong TCs is small, it does not mean that strong TCs do not exist, it is possible that in the training of the model, there have been cases where the intensity value reached 140 kt or even 150 kt, and the intensity of LAN has been steadily increasing over some time, so the model predicts that it is a sustained enhancement, i.e., it is assumed that 128 kt in the red box is not the peak value on this occasion.

However, for the intensification after the red box marking, the model predictions are more accurate, as shown by the blue box in Figure 8d. In the blue box, the model accurately predicts that LAN peaks. For the overall trends in DOKSURI and LAN, the model predicts them reasonably well, with prediction deviations within 10 kt. This confirms the good stability and accuracy of our model AWL-Net for 24 h TC intensity prediction.

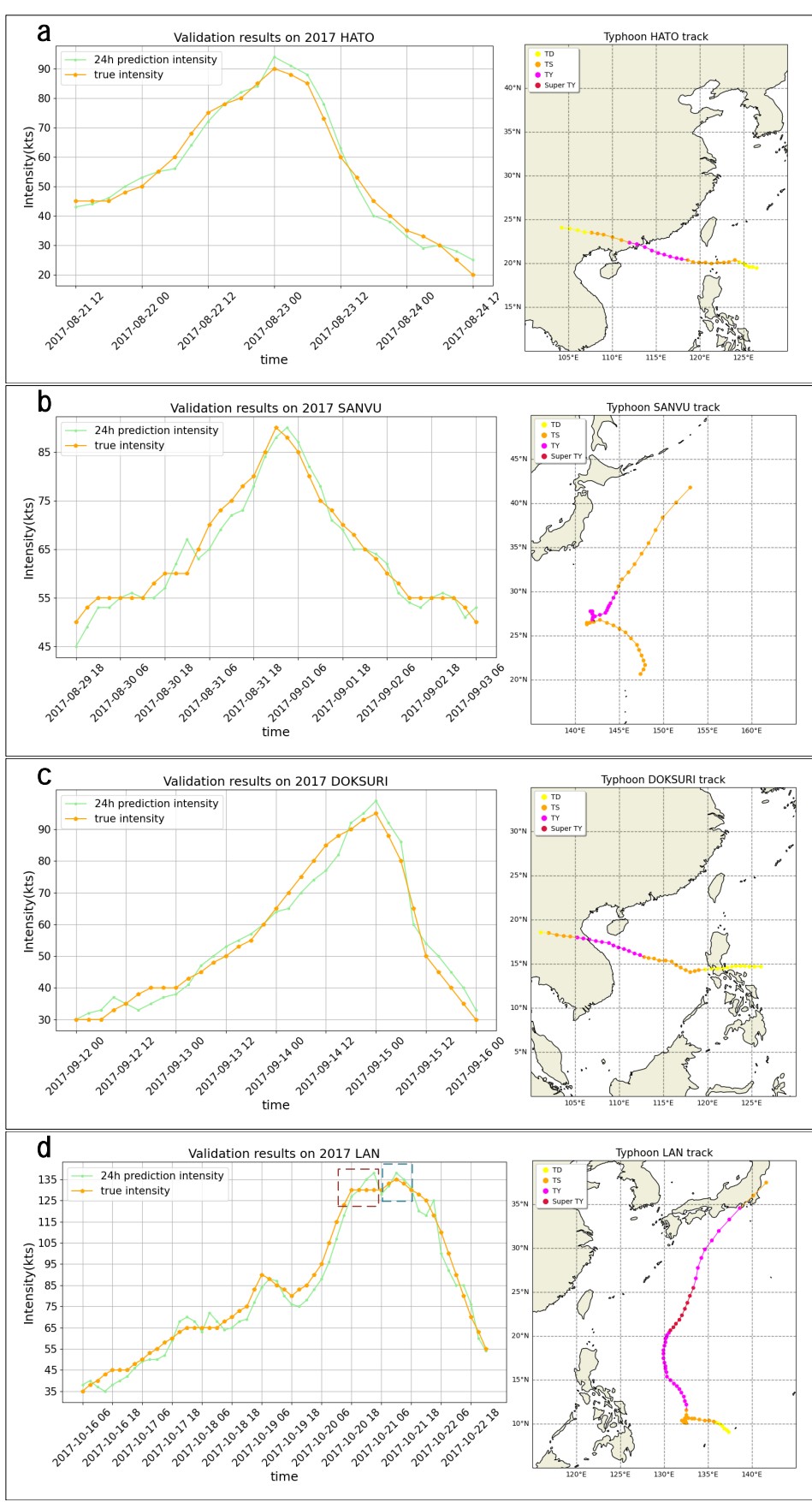

**Figure 8.** Visualization of the prediction performance of the AWL-Net and track. (**a**) TC HATO in 2017. (**b**) TC SANVU in 2017. (**c**) TC DOKSURI in 2017. (**d**) TC LAN in 2017.

## 5. Conclusions

In this study, a deep learning-based model AWL-Net for the short-term intensity prediction of TCs is proposed. Considering the environmental information around the TCs, such as cloud temperature distribution, water vapor content, etc., and the structural changes in the TCs, the symmetry of the convective structure of the clouds is reflected by the DAV, and high-dimensional features are efficiently downscaled to low-dimensional but rich information by the Hovmöller diagram. A double-branch structure is designed to construct an FE module to mine its spatio-temporal features and efficiently explore the relationship between multi-source data and targets. Second, the FF module integrates the multi-source data into a more comprehensive feature representation, which helps the model understand the data better and improves the predictive performance of the model. For the differences between multi-source data, AWL is used to autonomously adjust the weights of the double-branch, learn the mapping features of different data and targets, and improve the expression ability of the model.

TC intensity prediction studies are difficult to predict accurately due to their complex physical processes. We assess the validity and generalizability of the AWL-Net model through ablation experiments and comparison experiments with baseline. When compared with the baseline method, our model improves the prediction accuracy and has a better fit on single-case validation. For 24 h TC intensity prediction studies, there is a common problem of low prediction by strong TCs, and although our model predictions improve on this problem, it still exists. By analyzing the experiments of intensity change and TC classification prediction within 24 h, we learned that both sudden TC intensity change and strong TC affect the model's prediction performance. For TCs in RI, our predicted intensity changes are small, and how to improve the model's extraction of TC features when they are in RI becomes a key issue.

**Author Contributions:** P.S., data processing, investigation, methodology of the study, writing the manuscript. W.T., funding acquisition, study management and supervision, and serves as the corresponding author. Y.C. and H.X., investigation. C.J., formal analysis. K.T.C.L.K.S., formal analysis, review, and writing and editing. All authors have read and agreed to the published version of the manuscript.

**Funding:** This research was jointly supported by the National Natural Science Foundation of China (Grants 42375147 and Grants 42075138) and the Program on Key Basic Research Project of Jiangsu (BE2023829).

**Data Availability Statement:** Data are contained within the article.

**Conflicts of Interest:** The authors declare no conflicts of interest.

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
