# Peer review of "Short-Term Intensity Prediction of Tropical Cyclones Based on Multi-Source Data Fusion with Adaptive Weight Learning"

_remotesensing, doi:10.3390/rs16060984_

Round 1
Reviewer 1 Report
Comments and Suggestions for Authors
The overall quality of the paper is good, with numerous reviews and analyses of relevant research. The proposed model also has good innovation.
But there are mainly several issues that need improvement.
1. The research data is not new enough, it is recommended to increase the data to 2023.
2. The basis for setting the parameters in the section 2.3 should be explained.
3. For the "CMA、JTWC and NHC" methods, these three models should be briefly introduced to improve the readability of the paper.
4. For the model proposed in this paper, improving accuracy alone is not enough. The performance of a model also depends on its operating cost. Suggest studying and analyzing the time and spatial complexity of the model proposed by the author compared to the comparative model. Especially the operational efficiency or speed of each model.
5. There are not enough cases, only 2 cases. And the representativeness of the cases are not strong enough. Suggest adding at least one more case.
Detailed review comments can be found in the annotations in the paper review draft.
Suggest reviewing after major modifications.

Reviewer 2 Report
Comments and Suggestions for Authors
Dear Authors,
Thank you for submitting your research regarding tropical cyclone (TC) tracking through satellite imagery. The topic is important and needs more research as submitted, but I found some issues that should be solved.
The first one is English: although the text is not badly written, there are many word repetitions that complicate the readability of the document.
The second point is that I was not able to find the main goal of the research. The Authors expose how they have made their research at the end of the Introduction, but the objective is not present.
More comments:
Avoid using "DAV" in the keywords
Table 1: What is PMW? It is true that the definition appears in the text, but it should also included in the caption.
L130-132: Were VIS channels used in the original research (Chen et al.)?
L134: What is GridSAT?
L138: What is CMORPH?
L145: How do you define "the best track provided by the Joint Typhoon Warning Centre (JTWC)"?
"The grey area block in Figure 2" I cannot see any grey area.
From Table 3, it seems that your proposal does not improve other methodologies. I am right?
Add the category's names in Table 7.
Comments on the Quality of English LanguageThere are many word repetitions that complicate the readability of the document.
Reviewer 3 Report
Comments and Suggestions for Authors
The manuscript is well done. I just have a couple of suggestions before publication.
The first concerns the addition of a description of the most important Tropical Cyclones that occurred in the past in the Introduction section. Instead, the other suggestion concerns moving the parts of discussion present in section 3 "TC Intensity Prediction and Evaluation" to section 4 "Analysis and Discussion".
Reviewer 4 Report
Comments and Suggestions for Authors
The authors conducted a forecast test for the maximum wind speed of tropical cyclones in the western North Pacific in 24 hours by constructing a deep-learning model AWL-Net and then training and validating it with the TCIR dataset. The results yielded MAE of 8.07 kt and RSME of 10.62 kt during 2014-2017. The MAE exceeded the forecast by the incumbent agency.
This paper is carefully written, both in the introduction and in the data and methods. The results are also well written. However, the comparison is limited to the combination of data used and the forecast errors of the incumbent agencies. I think it necessary of the comparison to other deep learning modules. Therefore, as a reviewer, it is not possible to determine whether the AWL-Net method proposed in this study is superior to other deep-learning models.
There are various phases of tropical cyclones from the genesis to extratropical cyclones. IN the current manuscript, the authors divided tropical cyclones by intensity classes as shown in Figure 7a and then showed a histogram on the number of tropical cyclones in each category, but it is not clear whether they are actually showing the frequency of data in the JTWC or CMA best track because it depends on TCIR dataset. Even within the same tropical cyclone class, the brightness temperature distribution shown by satellite observations is likely to be different, as it includes intensification, mature, and decaying phases. How does this study treat phenomena in which the horizontal scale of the disturbance becomes larger, for example, extratropical transition? The comment is relevant to the content of L240.
I have an impression that this study has succeeded in including symmetry of a tropical cyclone in the authors' deep-learning model by using DAV. However, the accuracy for rapid intensification is still not good. It will be interesting to see if other deep learning methods can show similar results for rapid intensification.
Figure 7b shows that there are predictions with extraordinary positive bias in relatively weak intensity classes. I am also interested to see if other deep learning methods show similar results in this regard, although the impact on MAE may be small.
In fact, I believe that the following major revisions would be necessary to publish this paper in Remote Sensing.
1. The TCIR dataset should be described in detail. This will give readers information particularly on spatial resolution and kind of satellites.
2. Again, proof that AWL-Net is significantly superior to other deep learning methods is required.
3. For 1D data, there is a strong correlation between intensity and R35. Also, in 2017, there is a strong correlation between DAV and intensity, R35. Is there any intention to include such data in 1D DATA? In other words, are the differences in the correlations in the 1-D data related to the forecast error between 2015 and 2017 tropical cyclone seasons?
Also, what is the difference between the intensity archived in the 1D data and the intensity in the 24-hour intensity forecast? Is 1D data included in CLIPER information?
Minor comments
a. Please spell out the following LSTM, RNN, CMA, NHC and so on.
b. Does the NHC forecast tropical cyclones in the western North Pacific?
c. What do the figures in Table 6 indicate and what is the analysis period?
d. Information on the tracks of two tropical cyclones is needed in Figure 8. Also, there is a lot of public interest in Typhoon HATO in 2017. Please add this as a case study if possible.
Round 2
Reviewer 1 Report
Comments and Suggestions for Authors
The author revised the paper well, and basically revised most of the questions raised by the reviewers. For the reviewer's reply, it is basically possible.
However, it is very bad for the author to modify and answer the time complexity and space complexity of the model.
Because most of the authors are researchers from the Institute of Software, they should not be unaware of the importance of the time complexity of studying algorithms or models. The time complexity of the model should not only give its running time, but also discuss the level of its time complexity theoretically. It should be compared, analyzed and discussed with the models involved in the paper.
With the spatial complexity, the authors should not only give the memory footprint of the model. The quantity and magnitude of variables should be analyzed and discussed, as well as compared with other models involved.
Therefore, it is suggested that the authors make a major revision again and review it again.
Reviewer 4 Report
Comments and Suggestions for Authors
I appreciate that the authors incorporate my previous comments into the revised manuscript. The quality of the paper seems to have improved. However, I would like to confirm essential points. Therefore, major revision is recommended before publication.
Major comments
1. TCIR uses data from JTWC and HURDAT2. This indicates that the maximum sustained wind speed predicted in this study is an 1-minute average. However, this study also uses the maximum sustained wind speed and the TC intensity clarification (Table 7) based on the CMA best track data, although the maximum sustained wind speed is a 2-minute average (or 10-minute?). In tropical cyclone research communities, JTWC and CMA best-track maximum sustained wind speeds are known to be significantly different. In other words, the reason for the clearly higher improving rate against CMA in L419 could be that the definition of maximum sustained wind speed is different in the first place. I can also understand why the sample size for super typhoon in Figure 7 is so large. If this paper is published as is, the case analysis and classifications may be better aligned with JTWC as long as TCIR is used, as it is likely to receive significant criticism from the tropical cyclone research community.
2. Passive microwave rain rate is only related to the amount of precipitation, and precipitation means that it is accompanied by downdrafts. Convection, on the other hand, involves updraft. If the authors want to see convection properly, the authors should use the polarization-corrected temperature (PCT) in the 89-91 GHz band.
Minor comments
3. There is no description regarding the angle of 'ROTATE' in Figure 1. Please provide a description of how the angle of rotation is determined.
4. In Figure 6c, please align the vertical scale with the horizontal one.
5. L340: Please spell out the full spelling of "Hov".
